# Detection of Polymorphisms in *FASN*, *DGAT1*, and *PPARGC1A* Genes and Their Association with Milk Yield and Composition Traits in River Buffalo of Bangladesh

**DOI:** 10.3390/ani14131945

**Published:** 2024-06-30

**Authors:** Monira Akter Mou, Gautam Kumar Deb, Md. Forhad Ahmed Hridoy, Md. Ashadul Alam, Hasi Rani Barai, Md Azizul Haque, Mohammad Shamsul Alam Bhuiyan

**Affiliations:** 1Department of Animal Breeding and Genetics, Bangladesh Agricultural University, Mymensingh 2202, Bangladesh; monira.mou1705@gmail.com (M.A.M.); forhad44911@bau.edu.bd (M.F.A.H.); 2Buffalo Production Research Division, Bangladesh Livestock Research Institute, Dhaka 1341, Bangladesh; debgk2003@blri.gov.bd (G.K.D.); ashadul@blri.gov.bd (M.A.A.); 3School of Mechanical and IT Engineering, Yeungnam University, Gyeongsan 38541, Gyeongbuk, Republic of Korea; hrbarai@ynu.ac.kr; 4Department of Biotechnology, Yeungnam University, Gyeongsan 38541, Gyeongbuk, Republic of Korea

**Keywords:** candidate gene, genotype, milk properties, genetic association, SNP

## Abstract

**Simple Summary:**

Riverine buffalo are a vital component of the agricultural economy in Bangladesh, contributing significantly to milk production and the livelihoods of many farmers. These animals are highly valued for their ability to thrive in local conditions and their substantial milk yield, which is important for meeting the dietary needs of the population. Despite their importance, efforts to enhance their productivity through genetic improvements have been limited. This study investigated single-nucleotide polymorphisms (SNPs) in the fatty acid synthase (*FASN*), diacylglycerol o-acyltransferase 1 (*DGAT1*), and peroxisome proliferator-activated receptor-γ coactivator-1α (*PPARGC1A*) genes to determine their association with milk yield and composition traits in riverine buffalo. The objective was to identify genetic markers for selective breeding to enhance milk production and quality. Several SNPs were identified in these genes, showing significant associations with key traits such as daily milk yield (DMY), fat percentage, protein percentage, and solids-not-fat percentage (SNF%). These results suggest that the identified polymorphisms could serve as valuable molecular markers for breeding programs aimed at improving milk yield and composition in riverine buffalo, offering a promising approach to boost the dairy industry in Bangladesh while contributing to food security and economic development within the region.

**Abstract:**

This study aimed to identify SNPs in the intron, exon, and UTR regions of the *FASN*, *DGAT1*, and *PPARGC1A* genes and to investigate their possible association with milk yield and composition traits in the riverine buffalo of Bangladesh. A total of 150 DNA samples from riverine buffalo were used for PCR amplification with five pairs of primers, followed by association studies using a generalized linear model in R. SNP genotyping was performed by direct sequencing of the respective amplicon. Traits analyzed included DMY, fat%, protein%, and SNF%. This study identified 8 SNPs in *FASN* (g.7163G>A and g.7271C>T), *DGAT1* (g.7809C>T and g.8525C>T) and *PPARGC1A* (g.387642C>T, g.387758A>G, g.409354A>G, and g.409452G>A). Genotypic and allelic frequencies differed significantly for each SNP genotype and did not follow the Hardy–Weinberg principle (*p <* 0.01 or *p <* 0.001) in most cases. The g.7163G>A and g.7271C>T SNP genotypes of the *FASN* gene were significantly associated with milk fat%, with the latter also significantly associated with SNF%. The g.8525C>T polymorphism of the *DGAT1* gene significantly affected protein% (*p <* 0.01). Additionally, *PPARGC1A* gene polymorphisms showed significant associations: g.387642C>T with fat% (*p <* 0.05); g.387758A>G and g.409354A>G with protein% (*p <* 0.001) and SNF% (*p <* 0.01); and g.409452G>A with DMY (*p <* 0.001), fat% (*p <* 0.05), and protein% (*p <* 0.01). Reconstructed haplotypes of the *PPARGC1A* gene were significantly associated (*p <* 0.01) with all traits except SNF%. These findings suggest that polymorphisms in these three candidate genes have the potential as molecular markers for improving milk yield and composition traits in the riverine buffalo of Bangladesh.

## 1. Introduction

The water buffalo (*Bubalus bubalis*) holds a significant position in agricultural landscapes globally, particularly in regions like Southern Asia, South America, Southern Europe, and Northern Africa. Globally, their population is estimated at around 158 million, with a staggering 97% inhabiting tropical and subtropical areas of Asia [1]. With its robustness, adaptability, and substantial contributions to milk and meat production, the water buffalo has long been revered as the “Black Gold” of South Asia [2]. These animals are characterized by their large size, curved horns, and distinctive hump, and they thrive in various climates, from humid tropical regions to arid grasslands. Water buffalo has two distinct sub-populations that can be differentiated through chromosome number. The river type has 50 chromosomes, whereas the swamp type possesses 48 chromosomes. In Bangladesh, the water buffalo stands as an indispensable pillar of agricultural livelihoods and food security, embodying a profound cultural and economic significance. In the context of Bangladesh, where agriculture remains the backbone of the economy and a primary source of livelihood for a vast majority of the population, the buffalo emerges as a pivotal asset, particularly in the dairy industry [3]. Its robustness enables it to thrive in diverse ecological niches, from the fertile plains of the Ganges Delta to the marshlands of the Sundarbans. Unlike cattle, buffaloes are well suited to the low-lying, waterlogged terrain prevalent in many parts of the country, making them invaluable allies for farmers navigating challenging environmental conditions [4]. Moreover, the buffalo plays a pivotal role in addressing food security and nutritional needs, especially in rural areas where access to alternative sources of protein and essential nutrients may be limited [5]. Beyond its role in dairy production, the buffalo’s draft power contributes significantly to agricultural productivity, facilitating plowing, transportation, and other farm-related activities. This dual-purpose functionality underscores its multifaceted utility and enduring importance in Bangladesh’s agrarian landscape.

Buffalo milk’s position as the second most consumed dairy product globally underscores its widespread appeal and nutritional value. Compared to cow’s milk, buffalo milk boasts superior qualities, including higher levels of fat, protein, lactose, vitamins, and minerals [6]. This nutritional richness, coupled with its distinctively creamy consistency and white color, not only enhances its flavor but also makes it a preferred choice for a wide range of culinary applications. The versatility of buffalo milk extends beyond its use as a beverage. Its thicker texture and higher fat content make it an ideal ingredient for producing an array of fat-based dairy products, each renowned for its indulgent taste and texture. From rich and flavorful butter to aromatic ghee, tangy yogurt, artisanal cheeses, and luxurious ice cream, buffalo milk serves as the foundation for a diverse and culturally significant range of dairy delicacies enjoyed by consumers worldwide [7,8]. Moreover, the burgeoning awareness among consumers of the numerous health benefits associated with buffalo milk further solidifies its appeal. Studies have shown that buffalo milk tends to have lower cholesterol levels compared to cow’s milk, making it a favorable choice for individuals seeking heart-healthy options [9]. Additionally, buffalo milk is rich in bioactive compounds and antioxidants, which are believed to confer various health-promoting properties, including immune support and disease prevention [10].

The composition of buffalo milk, however, is not static and can be influenced by modifications in the feeding regimen of dairy buffaloes, resulting in temporary alterations in milk fat and protein content [11]. Consequently, a strategic approach involves selecting animals with a known genetic architecture, which could potentially yield milk with superior composition [12]. Recent advancements in DNA genotyping technology have facilitated comprehensive investigations into the genetic basis of milk composition traits in buffalo. Previous studies have identified candidate genes responsible for milk yield and composition traits in buffaloes including *FASN*, *PPARGC1A*, and *DGAT1* [13]. Among these candidate genes, SNPs in *FASN*, *PPARGC1A*, and *DGAT1* significantly influence milk fat content [14,15], protein percentage [9], and fatty acid compositions [16] in different buffalo populations. 

The *FASN* gene is responsible for encoding the fatty acid synthase enzyme, which plays a crucial role in catalyzing the synthesis of long-chain saturated fatty acids. Notably, specific SNP genotypes within the *FASN* gene, such as g.7164G>A and g.8927T>C, have demonstrated significant associations with milk traits in Mediterranean buffalo populations [17]. Furthermore, variants in the exon regions of the *DGAT1* gene have been linked to a range of milk production traits across different buffalo breeds [18,19]. In buffalo, the *PPARGC1A* gene holds considerable importance in various biological processes related to milk production and quality. Specifically, polymorphisms within exon 8 and 13 of the *PPARGC1A* gene have shown significant associations with milk production traits in Italian Mediterranean buffalo populations [6]. 

These polymorphisms in candidate genes could have potential implications in marker-assisted selection (MAS) programs for further improving milk traits in the Bangladeshi riverine buffalo population. Despite the significant impacts of SNP-specific association studies on genetic improvement experimentation in various livestock species, there have been very few studies conducted in Bangladesh, particularly regarding buffalo. Therefore, this study aims to identify SNPs in selected exons and their adjacent regions of the three aforementioned genes and investigate their possible associations with milk yield and composition traits in the riverine buffalo of Bangladesh. Moreover, this research not only contributes to the scientific understanding of buffalo genetics but also offers practical implications for breeding programs aimed at enhancing milk production efficiency and quality. By identifying SNPs associated with desirable milk traits, such as fat content, protein percentage, and fatty acid compositions, breeders can make informed decisions in selecting animals with superior genetic profiles, thereby accelerating genetic improvement efforts. 

## 2. Methods and Materials

### 2.1. Animals and Phenotypes

A total of 150 lactating river-type buffaloes were selected for this study from six different populations: Bangladesh Livestock Research Institute, Savar, Dhaka (n =16); Godagari, Rajshahi (n = 18); Ishwardi, Pabna (n = 22); Madarganj, Jamalpur (n = 12); Companiganj, Noakhali (n = 34); and Buffalo Breeding and Development Farm, Bagerhat (n =48) (Figure 1). Farmers in these regions have a long tradition of buffalo husbandry due to geographic advantages, such as river basins and coastal regions, resulting in the predominance of indigenous river-type buffalo in the selected areas. The genotype of the selected animals was confirmed by tracing back pedigree information and engaging with farmers.

Herdbook-based record-keeping was used to produce objective data on milk production and composition by designated personnel through regular visits to the respective farms or populations. The International Committee for Animal Registration (ICAR) guidelines were adhered to for buffalo milk recording. Specifically, milk composition data (SNF, fat, and protein percentages) were obtained using a portable Lactoscan machine (Lactoscan Milk analyzer, Model 1010, Farm Eco, Nova Zagora, Bulgaria), and at least three consecutive records were averaged to obtain the final value. The traits considered for this study included daily milk yield in liters (DMY), milk fat percentage (%), protein percentage (%), and solids-not-fat (SNF) percentage (%). The details of phenotypic distribution information for the studied animals are presented in Appendix A.

### 2.2. Blood Sampling and DNA Extraction

Approximately 5.0 mL blood samples were aseptically collected from the selected animals using venoject tubes coated with EDTA as an anticoagulant. Genomic DNA was extracted from the whole blood samples using the AddPrep Genomic DNA Extraction Kit (ADD BIO INC., Daejeon, Republic of Korea), following the manufacturer’s instructions with slight modifications to the protocol, adding RBC lysis buffer at the beginning of the DNA extraction process. The concentration and purity of the isolated genomic DNA were assessed using a NanoDrop spectrophotometer (Model ND2000, Thermo Fisher Scientific, Wilmington, NC, USA).

### 2.3. PCR Amplification

Five pairs of primers were selected from the previous studies conducted by Ye et al. [17], Yuan et al. [20], and Hosseini et al. [6] to amplify exon 10 of *FASN*, exon 13 and 17 of *DGAT1*, and exon 8 and 13 (UTR) of *PPARGC1A* genes. These specific amplicons were found to be associated with milk composition traits across various buffalo populations. Primer synthesis was performed by a commercial service provider (Macrogen Inc., Seoul, Republic of Korea). PCR amplification was conducted in a 20 µL reaction volume comprising 1.5 µL DNA, 10 µL of 2× master mix (Prime Taq DNA polymerase 1 unit/10 µL, 20 mM Tris-HCl (pH-8.8), 100 mM KCl, 0.2% Triton^®^ X-100, 4.0 mM MgCl_2_, enzyme stabilizer, sediment, loading dye, and 0.5 mM each of dNTP), 2.0 μL of each primer (10 pmol/μL), and 4.5 μL deionized water (ADD BIO INC., Daejeon, Republic of Korea) using a TGradient Thermocycler (Biometra, Göttingen, Germany). The thermal profile consisted of initial denaturation at 95 °C for 10 min followed by 30–35 cycles with denaturation temperature set at 95 °C for 30 s, annealing at 60–62 °C for 45 s, extension at 72 °C for 45 s, and a final extension at 72 °C for 10 min. The resulting PCR product was electrophoresed in a 2% agarose gel stained with green gel dye and visualized using a digital gel documentation system (GDS-200, Sunil-Bio Inc., Seoul, Republic of Korea).

### 2.4. Sequencing and Polymorphism Detection

A subset of pooled DNA samples was initially used for amplifying each gene fragment, and the purified PCR products were sequenced in both directions by a commercial sequence service provider (Wuhan Tianyi Huayu Gene Technology Co., Ltd., Wuhan, China). The resulting raw sequences were obtained using Chromas software (Version 2.6.6, Technelysium Pty. Ltd., South Brisbane, Australia). Multiple sequence alignment was conducted, incorporating reference sequences of *Bubalus bubalis FASN* (NC_059159.1), *DGAT1* (NC_059171.1), and *PPARGC1A* (NC_059163.1), using the CLUSTALW program [21] to identify polymorphisms in the sequenced fragments. Subsequently, all five pairs of primers were utilized for PCR amplification and Sanger sequencing of the amplified fragments. SNP genotyping of the five fragments of *FASN*, *DGAT1*, and *PPARGC1A* genes was carried out using 145, 82, 148, 144, and 148 sequence data, respectively. 

### 2.5. Statistical Analysis

Genotypic and allelic frequencies of the identified SNPs were calculated according to the methods outlined by Falconer and Mackay [22]. The haplotype reconstruction was performed using DnaSP v.6.12 software [23], and their association analysis was performed in the R platform. Single-marker association analysis, assessing the relationship between the resulting SNP genotypes of the *FASN*, *DGAT1*, and *PPARGC1A* genes and milk production traits, was conducted using the Agricole package in R [24]. Mean separation was assessed using the pastecs package in R (Version 4.3.3) [24]. Due to a high frequency of missing data associated with days in milk and parity of animals, the statistical model encountered convergence issues, preventing the inclusion of these effects in the model. These effects are important to explain the variability of milk yield and composition traits, and thus potential biases could have been introduced in the results due to lacking information on them. The model used for milk traits was as follows:$$Y_{ijklmn}=\mu +G_{i}+L_{j}+M_{k}+{GL}_{l}+{GM}_{m}+{LM}_{n}+e_{ijklmn}$$
where $Y_{ijklmn}$ represents the dependent variable (productive traits); µ denotes the overall mean; $G_{i}$ represents the fixed effect of $i^{th}$ genotype; $L_{j}$ denotes the fixed effect of $j^{th}$ location; $M_{k}$ signifies the fixed effect of $k^{th}$ management system; ${GL}_{l}$ represents the interaction between the fixed effects of genotype and location; ${GM}_{m}$ indicates the interaction between the fixed effects genotype and management system; ${LM}_{n}$ represents the interaction between the fixed effect location and management system; and $e_{ijklmn}$ denotes the random error. The management system in the model refers to the consideration of two types: intensive and semi-intensive.

## 3. Results

### 3.1. Descriptive Statistics of Milk Yield and Milk Composition Traits of Riverine Buffalo

The descriptive statistics presented in Table 1 provide a comprehensive overview of the milk yield and milk composition traits observed in riverine buffaloes in Bangladesh. The DMY data were collected from 142 buffaloes, showing a range from 1.03 to 5.50 L per day, with an average yield of 2.78 ± 0.06 L. The standard deviation (SD) for DMY was 0.721 L, indicating moderate variability, and a coefficient of variation (CV) of 25.92% reflected a significant degree of diversity in milk production among the buffalo population. The analysis of milk composition traits included 116 observations. The milk fat percentage varied significantly, ranging from 3.69% to 11.24%, with a mean value of 8.34% and an SE of 0.17%. The SD for milk fat was 1.800, and the CV was 21.59%, indicating substantial variation in fat content among the buffaloes. Protein percentage ranged from 2.20% to 6.29%, with a mean of 3.64% and an SE of 0.06%. The SD for protein was 0.687, and the CV was 18.86%, showing less variability compared to milk fat. In addition, the SNF percentage ranged from 6.45% to 12.63%, with a mean of 9.41% and an SE of 0.10%. The SD for SNF was 1.097, and the CV was 11.66%, suggesting relatively less variation in SNF content compared to other milk composition traits.

### 3.2. Detection of the Polymorphisms

The analysis of multiple sequences aimed at exploring the genetic landscape within the selected amplicons of *FASN*, *DGAT1*, and *PPARGC1A* genes in the riverine buffalo populations of Bangladesh uncovered a diverse array of genetic variations, identifying a total of eight SNPs (Table 2 and Figure 2). Within the *FASN* gene, two distinct SNPs were observed. The first SNP was located within intron 9 (g.7163G>A), while the second SNP was situated in exon 10 (g.7271C>T). Transitioning to the *DGAT1* gene, two SNPs were confirmed. The first SNP (g.7809C>T) was pinpointed within exon 13, whereas the second SNP (g.8525C>T) was identified within exon 17. Additionally, the analysis revealed two SNPs within the *PPARGC1A* gene, each strategically positioned within functionally significant regions. The first SNP (g.387642C>T) was localized within exon 8, while the second SNP (g.387758A>G) was also found in exon 8. Furthermore, two SNPs were detected in the 3′-UTR region of the *PPARGC1A* gene, specifically at positions g.409354A>G and g.409452G>A, respectively.

### 3.3. Population Genetic Information for the Identified SNPs in Three Candidate Genes

The genotypic and allelic frequencies, observed (Ho) and expected (He) heterozygosity, and the chi-square test (χ^2^) values for SNPs identified in the *FASN*, *DGAT1*, and *PPARGC1A* genes are summarized in Table 3. All detected polymorphisms exhibited higher observed heterozygosity (Ho) compared to expected heterozygosity (He), except for the g.7809C>T polymorphism in the *DGAT1* gene. The results from the chi-square test indicated a significant deviation of the studied population from the Hardy–Weinberg equilibrium (HWE). Out of eight polymorphisms detected in *FASN*, *DGAT1*, and *PPARGC1A* genes, six of them (g.7163G>A, g.7271C>T, g.7809C>T, g.8525C>T, g.387758A>G, and g.409354A>G) possessed three genotypes for each mutation and the remaining g.387642C>T and g.409452G>A polymorphisms of the *PPARGC1A* gene had only two genotypes (Table 3).

### 3.4. Association between the SNPs of FASN and DGAT1 Genes with Milk Traits

The effects of identified SNP genotypes within the *FASN* and *DGAT1* genes on milk yield and milk composition traits are illustrated in Table 4. This study indicated significant effects (*p <* 0.05) of the g.7163G>A SNP genotypes of the *FASN* gene solely on fat%, whereas the g.7271C>T polymorphism showed highly significant effects (*p <* 0.01) on both fat% and SNF%. Specifically, the fat% varied across genotypes, with CC, CT, and TT genotypes (g.7271C>T) corresponding to 8.50 ± 0.24, 7.84 ± 0.27, and 9.44 ± 0.45, respectively. Conversely, these genotypes displayed contrasting effects on SNF production, with the CC genotype yielding significantly higher SNF (9.74 ± 0.12%) compared to the CT (9.47 ± 0.15%) and TT (8.63 ± 0.47) genotypes. Moreover, the g.8525C>T SNP genotypes of the *DGAT1* gene exhibited a highly significant association (*p <* 0.01) solely with milk protein%. Specifically, the TT and CT genotypes yielded significantly higher milk protein (3.86 ± 0.20 and 3.75 ± 0.09%) compared to the CC genotype (3.46 ± 0.09%).

### 3.5. Association between SNP Genotypes of PPARGC1A Gene and Milk Traits

The g.409452G>A polymorphism of the *PPARGC1A* gene exhibited a highly significant association (*p <* 0.0009) with the DMY trait (Table 5). Specifically, the homozygous GG genotype was associated with an increase of 0.44 L in milk yield compared to the heterozygous GA genotype. Milk fat% showed significant associations with the g.387642C>T (*p <* 0.043) and g.409452G>A (*p <* 0.031) SNP genotypes. In both cases, the heterozygous CT (7.81 ± 0.36%) and GA (7.64 ± 0.27%) genotypes produced 0.7 to 1.0% less fat compared to the homozygous CC and GG genotypes (8.52 ± 0.17 and 8.54 ± 0.20%). Notably, milk protein% showed a highly significant association with the g.387758A>G (*p <* 0.001), g.409354A>G (*p <* 0.0004), and g.409452G>A (*p <* 0.003) polymorphisms. Furthermore, highly significant associations were observed between the g.387758A>G (*p <* 0.018) and g.409354A>G (*p <* 0.002) polymorphisms and SNF%. The homozygous mutant GG genotypes were significantly associated with increased SNF%.

### 3.6. Association between Constructed Haplotypes of PPARGC1A Genes and Milk Traits

The association between the reconstructed haplotypes of the *PPARGC1A* gene milk traits is outlined in Table 6. A total of 11 different haplotypes were identified, all of which exhibited a highly significant association (*p <* 0.001) with DMY and milk composition traits, excluding SNF% (Table 6). Haplotype 1 (CAAG) had the highest frequency (0.26), while Haplotypes 5 (CAGA) and 11 (TAAA) resulted in comparatively lower frequency (0.02) than the others. Among all haplotypes, Haplotype 4 (CGGG) demonstrated the highest DMY (3.01 ± 0.10 L), and both Haplotypes 4 (CGGG) and 9 (TAAG) exhibited the highest protein percentages compared to the other haplotypes. Haplotype 2 (CAGG) recorded the highest milk fat percentage at 8.63 ± 0.24, while Haplotype 11 (TAAA) exhibited the highest SNF%. 

## 4. Discussion

Milk production represents a complex quantitative trait influenced by numerous genes. Therefore, identifying candidate genes and genetic markers associated with milk yield and composition traits is crucial for implementing marker-assisted selection (MAS) at a population level [12,25,26]. Previous studies utilizing quantitative trait locus (QTL) analysis and association studies have identified several genes, including *DGAT1*, *GHR*, *FASN*, and *PPARGC1A*, as promising candidates in buffalo for milk production traits [27,28]. 

The average milk yield of river-type buffalo in Bangladesh was determined to be 2.78 L/day in this study, which is comparatively higher than the values reported by Rahman et al. [29] and Samad et al. [30]. These studies reported average milk production ranging from 1.93 to 2.18 ± 0.63 L/day in the indigenous river-type buffalo of Bangladesh. The average milk fat% (8.34 ± 0.17%) observed in this study, ranging between 3.69% and 11.24%, aligns with the findings of Samad et al. [30], who reported milk fat% ranging from 7.2% to 9.1% in the riverine buffalo populations of Bangladesh. However, de Camargo et al. [31] and Rahman et al. [29] found relatively lower fat% in buffalo milk, ranging between 6.55% and 7.02%. The protein% and SNF% obtained in this study are consistent with the findings of Rahman et al. [29] and Samad et al. [30]. In the studied buffalo population, the standard deviation and coefficient of variation of milk fat and protein percentages were comparatively higher than those reported by de Camargo et al. [31], indicating a large variation within the studied population. Factors such as feeding regimes, age, stage of lactation, and animal genotype significantly influence milk yield and composition traits [13,32,33].

To date, several polymorphic sites have been identified in the bovine *FASN* gene. However, limited reports exist regarding *FASN* gene polymorphisms in buffalo [17,34]. In our study, we conducted a scan of exon 10 of the *FASN* gene in riverine buffalo, identifying two SNPs, g.7163G>A, and g.7271C>T, the latter of which was previously reported in Mediterranean buffalo [17]. Consistent with the findings of Li et al. [15], our study also identified two polymorphisms, g.7809C>T and g.8525C>T in exons 13 and 17 of the *DGAT1* gene. Additionally, the polymorphism reported in exon 17 of the *DGAT1* gene was also found in Chinese water buffalo and Murrah buffaloes [18,20,35], respectively, supporting our findings. We detected a total of four polymorphisms in the *PPARGC1A* gene, two in exon 8, and the remaining two SNPs in the 3′untranslated region (UTR). Hosseini et al. [6] reported one SNP in exon 8 and three SNPs in exon 13 in Italian Mediterranean buffaloes, partially corroborating our investigation. The higher observed heterozygosity (Ho) compared to expected heterozygosity (He) suggests that the studied buffalo populations exhibit greater genetic diversity than anticipated. However, a significant departure from the Hardy–Weinberg equilibrium (HWE) was observed for the identified polymorphisms. This discrepancy could stem from various factors, including selection pressure, interspecific hybridization, population substructure, and the demographic history of the buffalo population under study [36,37]. 

The g.7164G>A and g.7272T>C SNP genotypes showed significant association with 270-day peak milk yield (*p <* 0.05), protein% (*p <* 0.01), and fat% (*p <* 0.01) in Mediterranean buffalo [17,38]. Similarly, a highly significant association (*p <* 0.01) was detected between fat and SNF% for the g.7271C>T SNP genotypes in Bangladeshi riverine buffalo which was consistent with the study of Ye et al. [17]. Kumar et al. [39] reported associations between polymorphisms of exon-40 of the *FASN* gene and lactation fat%, lactation total solid average, and peak yield in Murrah buffaloes that support the present investigation. Furthermore, polymorphisms of the *FASN* gene were significantly associated with milk fat and protein content in dairy cows [40], aligning with our study. Given that the identified g.7271C>T polymorphism was significantly correlated with milk fat and SNF contents in the studied buffalo population key indicators of milk quality, this SNP holds promise as a molecular marker for selecting buffalo that produce high-quality milk.

Furthermore, the present study highlights a highly significant association (*p <* 0.01) between the g.8525C>T polymorphism of the *DGAT1* gene and protein% within the studied population. However, a significant association (*p <* 0.05) was found with fat% and protein% in Murrah buffaloes, partially aligning with the current findings [41]. Additionally, the same SNP (g.8525C>T) genotypes exhibited a significant association with fat% in riverine buffalo population [15]. Furthermore, they found a significant (*p <* 0.05) association between peak milk yield, total milk yield, and protein% with the g.8330T>C polymorphism located in exon 13, where the C variant was associated with increased milk yield but lower protein% compared with the T variant. Similar to the present study, Yuan et al. [20] found a non-significant association for the g.8525C>T polymorphism with milk production traits in Chinese water buffalo populations. Conversely, none of the milk production traits reached a significant level (*p* > 0.05) for the g.7809C>T polymorphism in exon 13 of the *DGAT1* gene in riverine buffalo of Bangladesh, where the TT genotype exhibited higher fat, protein, and SNF percentages compared to the CC genotype. However, Cardoso et al. [42] reported a significant association of a variable number of tandem repeats (VNTR) in the promoter region of *DGAT1* with fat% in buffaloes.

The *PPARGC1A* gene serves as an inducible transcriptional coactivator implicated in the regulation of carbohydrate and fat metabolism in buffalo, potentially influencing milk fat synthesis [43]. Hosseini et al. [6] observed a significant correlation between the g.304050G>A and g.325997G>A polymorphisms and both milk yield and protein percentage in Italian Mediterranean buffalo, aligning in part with the findings of this study. Interestingly, Sihag et al. [44] identified significant variations in fat, protein, and SNF yield among AA, AB, and BB genotypes (SNP1-g.993A>T, SNP2-g.1237T>A, SNP3-g.1238G>C, and one insertion, g.1240_>G bp) in Murrah, Bhadawari, and Egyptian buffalo breeds, consistent with the current results. A recent study documented the impact of c.1598A>T and other SNPs in the *PPARGC1A* gene on lactation initiation and maintenance, milk yield, milk quality, and milk fat in Anatolian water buffaloes, supporting the findings of this investigation [45]. A total of 11 haplotypes were identified in the *PPARGC1A* gene of riverine buffalo in Bangladesh, mirroring the results of [43], who reported 10 haplotypes based on *PPARGC1A* gene polymorphisms. The more frequent haplotypes (Hap1, Hap2, Hap3, Hap4) exhibited superior milk yield and composition traits (fat and protein%) compared to less frequent haplotypes. Cobanoğlu and Ardicli [46] noted significant effects of combined genotypes of *PPARGC1A* and *LTF* gene polymorphisms in dairy cattle, with TTAB and TTAA genotypes demonstrating significantly higher milk fat content (5.86 ± 0.87%) and milk protein content (3.45 ± 0.07%), respectively, supporting the present study. El-Komy et al. [13] reported significantly enhanced milk performance (high milk yield, fat%, protein%, and 305-day milk, fat, and protein yield) associated with haplotypes of *GHR* gene polymorphisms in Egyptian buffaloes. Similarly, haplotypes of the *DGAT1* gene showed significant associations with milk yield, protein%, and fat% in buffalo [47], further corroborating the results of the haplotype-based association study in this research.

Genetic polymorphisms within these genes were found to have significant associations with key milk traits, such as daily milk yield, fat%, protein%, and SNF% content. The findings demonstrate that specific SNPs in these genes can serve as valuable genetic markers for selective breeding programs. By identifying and utilizing these markers, breeders can enhance the efficiency and effectiveness of breeding strategies aimed at improving milk production and quality in buffalo populations. In the context of Bangladesh, where dairy farming is a critical component of the agricultural sector, the insights gained from this study are particularly valuable.

## 5. Conclusions

This study is the first to explore *FASN*, *DGAT1*, and *PPARGC1A* gene polymorphisms and their association with milk traits in the riverine buffalo population of Bangladesh. Sequence analysis revealed the existence of eight polymorphisms in the intron, exon, and UTR regions of selected amplicons of these three candidate genes. The significant associations found in this study suggest that these polymorphisms have the potential to be used as molecular markers for improving milk traits in the riverine buffalo of Bangladesh. This could lead to more targeted and efficient breeding programs aimed at enhancing milk yield and composition, ultimately benefiting the dairy industry in the region.

## Figures and Tables

**Figure 1 animals-14-01945-f001:**
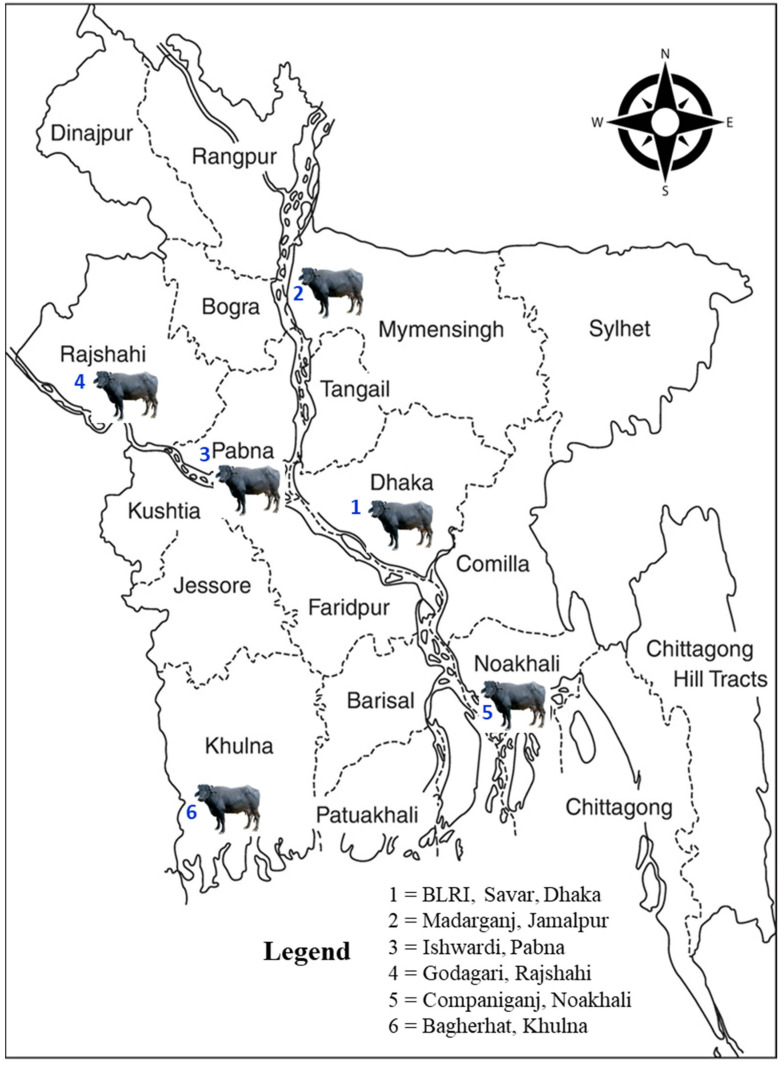
Map showing the locations of blood sampling and phenotypic performance data collection.

**Figure 2 animals-14-01945-f002:**
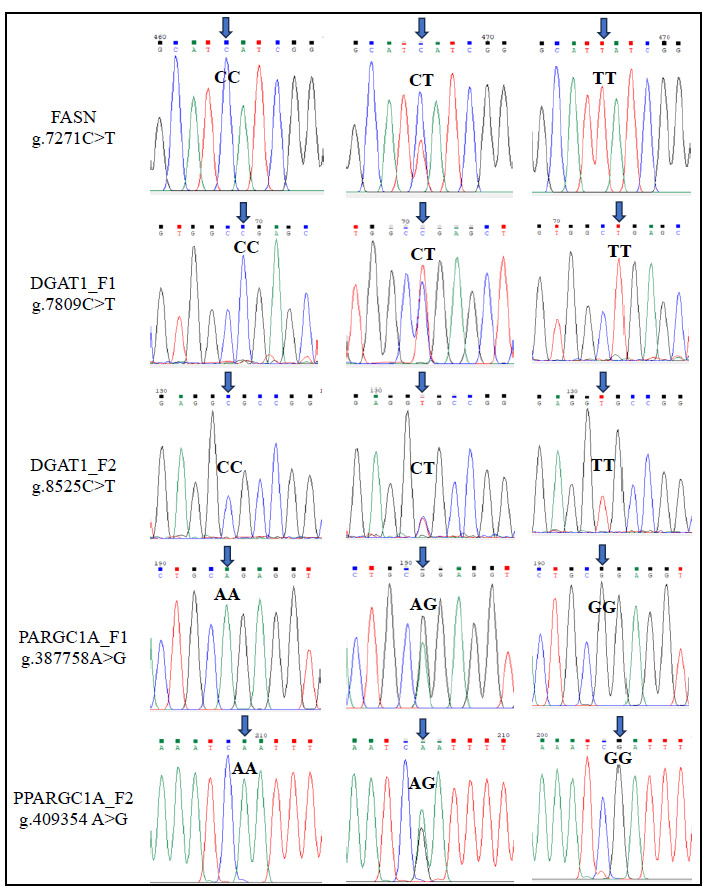
Partial sequences of five selected genotypes of the identified SNPs in *FASN*, *DGAT1*, and *PPARGC1A* genes.

**Table 1 animals-14-01945-t001:** Descriptive statistics of milk yield and milk composition traits in riverine buffalo of Bangladesh.

Trait	N	Minimum	Maximum	Mean ± SE	SD	CV%
DMY (liter)	142	1.03	5.50	2.78 ± 0.06	0.721	25.92
Milk fat%	116	3.69	11.24	8.34 ± 0.17	1.800	21.59
Protein%	116	2.20	6.29	3.64 ± 0.06	0.687	18.86
SNF%	116	6.45	12.63	9.41 ± 0.10	1.097	11.66

N, number of individuals; SE, standard error; SD, standard deviation; CV, coefficient of variation; DMY, daily milk yield; SNF, solids-not-fat.

**Table 2 animals-14-01945-t002:** Primer sequence information and identified SNPs in *FASN*, *DGAT1*, and *PPARGC1A* genes in river buffalo of Bangladesh.

Primer Set	Primer Sequence (5′ to 3′)	Product Size (bp)	Identified SNP	SNP Location
FASNF1 FASNR1	F: CCCACTCTGGTTCATCTGCTC R: CCTCCCACGAAGACCCTCA	660	g.7163G>A g.7271C>T	Intron 9 Exon 10
DGAT1F1 DGAT1R1	F: GCTGTTCTGGCACCTGGCAC R: CACCCACCTGATGCACCACT	300	g.7809C>T	Exon 13
DGAT1F2 DGAT1R2	F: AGGCTCACTCCCGTCCCAT R: GTGAGGCAAAGCAGTCCAAC	230	g.8525C>T	Exon 17
PPARGC1AF1 PPARGC1AR1	F: AGTGGACACGAGGAAAGGAAG R: GGGTGGGTTTTGACAAGGTT	724	g.387642C>T g.387758A>G	Exon 8
PPARGC1AF2 PPARGC1AR2	F: TGAACACATGCACCCCATCAT R: CGTGCCAGGAGTTTGGTTGT	789	g.409354A>G g.409452G>A	3′ UTR

SNP position is based on the reference sequences of *Bubalus bubalis. FASN* (NC_059159.1), *DGAT1* (NC_059171.1), and *PPARGC1A* (NC_059163.1).

**Table 3 animals-14-01945-t003:** Summary of population genetic information for the identified SNPs in *FASN*, *DGAT1*, and *PPARGC1A* genes in river buffalo.

Gene	SNP ^1^	Genotype Frequency ^2^			Allele Frequency		Heterozygosity		χ^2^ (*p*-Value)
							Ho	He	
*FASN*	g.7163G>A	GG	GA	AA	G	A	0.43	0.41	55.51 ***
		0.50 (72)	0.43 (63)	0.07 (10)	0.71	0.29			
	g.7271C>T	CC	CT	TT	C	T	0.42	0.39	67.43 ***
		0.52 (76)	0.42 (61)	0.06 (08)	0.73	0.27			
*DGAT1*	g.7809C>T	CC	CT	TT	C	T	0.46	0.47	10.20 **
		0.39 (32)	0.46 (38)	0.15 (12)	0.62	0.38			
	g.8525C>T	CC	CT	TT	C	T	0.54	0.48	11.57 **
		0.32 (48)	0.54 (80)	0.14 (20)	0.59	0.41			
*PPARGC1A*	g.387642C>T	CC	CT	TT	C	T	0.17	0.15	228.00 ***
		0.83 (120)	0.17 (24)	0.00 (00)	0.92	0.08			
	g.387758A>G	AA	AG	GG	A	G	0.52	0.47	19.26 ***
		0.37 (53)	0.52 (75)	0.11 (16)	0.63	0.37			
	g.409354A>G	AA	AG	GG	A	G	0.41	0.40	51.00 ***
		0.52 (61)	0.42 (49)	0.07 (08)	0.72	0.28			
	g.409452G>A	GG	GA	AA	G	A	0.21	0.18	161.25 ***
		0.79 (95)	0.21 (25)	0.00 (00)	0.90	0.10			

^1^ SNP position is based on the reference sequences of the *FASN* (NC_059159.1), *DGAT1* (NC_059171.1), and *PPARGC1A* (NC_059163.1) genes of *Bubalus bubalis*. ^2^ Values in the parentheses represent the number of samples in the respective SNP. ***, level of significance at *p <* 0.001; **, level of significance at *p <* 0.01.

**Table 4 animals-14-01945-t004:** Association between identified SNP genotypes of *FASN* and *DGAT1* genes and milk traits in river buffalo of Bangladesh.

Gene and SNP	Genotype	DMY (Liter)	Fat%	Protein%	SNF%
*FASN* g.7163G>A	GG	2.92 ± 0.08 (69)	8.57 ± 0.25 ^a^ (58)	3.74 ± 0.09 (58)	9.48 ± 0.14 (58)
	GA	2.73 ± 0.11 (61)	7.88 ± 0.25 ^b^ (47)	4.53 ± 0.75 (47)	9.48 ± 0.15 (47)
	AA	2.95 ± 0.12 (10)	8.80 ± 0.59 ^a^ (10)	3.65 ± 0.38 (10)	8.93 ± 0.42 (10)
*p* value		0.2198	0.0452	0.8260	0.2900
*FASN* g.7271C>T	CC	2.91 ± 0.07 (73)	8.50 ± 0.24 ^ab^ (62)	3.74 ± 0.09 (62)	9.74 ± 0.12 ^a^ (56)
	CT	2.75 ± 0.12 (59)	7.84 ± 0.27 ^b^ (45)	4.57 ± 0.78 (45)	9.47 ± 0.15 ^ab^ (45)
	TT	2.99 ± 0.21 (8)	9.44 ± 0.45 ^a^ (8)	3.67 ± 0.48 (8)	8.63 ± 0.47 ^b^ (8)
*p* value		0.2671	0.0099	0.8270	0.0069
*DGAT1* g.7809C>T	CC	2.90 ± 0.14 (35)	8.41 ± 0.38 (27)	3.80 ± 0.13 (27)	9.56 ± 0.20 (27)
	CT	2.70 ± 0.12 (29)	8.41 ± 0.37 (22)	3.96 ± 0.29 (22)	9.38 ± 0.14 (22)
	TT	3.01 ± 0.3 (10)	8.58 ± 0.64 (9)	3.70 ± 0.22 (9)	9.62 ± 0.31 (9)
*p* value		0.4133	0.9540	0.6785	0.6810
*DGAT1* g.8525C>T	CC	2.78 ± 0.10 (46)	8.37 ± 0.25 (39)	3.46 ± 0.09 ^b^ (39)	9.22 ± 0.17 (39)
	CT	2.78 ± 0.08 (78)	8.23 ± 0.24 (64)	3.75 ± 0.09 ^a^ (62)	9.45 ± 0.14 (64)
	TT	2.77 ± 0.18 (18)	8.75 ± 0.58 (13)	3.86 ± 0.20 ^a^ (13)	9.97 ± 0.34 (13)
*p* value		0.9980	0.5220	0.0056	0.2255

The different superscripts within the same column differ significantly at *p <* 0.01 and *p <* 0.05. The values in the parentheses indicate the number of observations per SNP genotype.

**Table 5 animals-14-01945-t005:** Association between identified SNP genotypes of *PPARGC1A* gene with milk traits in river buffalo of Bangladesh.

SNP	Genotype	DMY (Liter)	Fat%	Protein%	SNF%
g.387642C>T	CC	2.90 ± 0.07 (117)	8.52 ± 0.17 ^a^ (94)	3.75 ± 0.10 (97)	9.37 ± 0.11 (97)
	CT	2.66 ± 0.16 (21)	7.81 ± 0.36 ^b^ (19)	3.79 ± 0.16 (19)	9.55 ± 0.23 (19)
*p* value		0.1390	0.0434	0.7940	0.4830
g.387758A>G	AA	2.84 ± 0.12 (51)	8.26 ± 0.31 (42)	3.49 ± 0.08 ^b^ (42)	9.50 ± 0.17 ^ab^ (42)
	AG	2.81 ± 0.09 (71)	8.37 ± 0.21 (59)	3.82 ± 0.14 ^ab^ (59)	9.18 ± 0.13 ^b^ (59)
	GG	3.15 ± 0.17 (16)	8.06 ± 0.51 (15)	4.26 ± 0.25 ^a^ (15)	9.96 ± 0.33 ^a^ (15)
*p* value		0.1660	0.7563	0.0010	0.0185
g.409354A>G	AA	2.79 ± 0.09 (57)	8.42 ± 0.26 (52)	3.56 ± 0.08 ^b^ (51)	9.46 ± 0.14 ^ab^ (51)
	AG	2.97 ± 0.10 (49)	8.44 ± 0.24 (41)	3.87 ± 0.19 ^b^ (41)	9.00 ± 0.16 ^b^ (41)
	GG	3.06 ± 0.33 (8)	7.49 ± 0.87 (7)	4.64 ± 0.32 ^a^ (7)	10.26 ± 0.44 ^a^ (7)
*p* value		0.2821	0.2920	0.0004	0.0023
g.409452G>A	GG	2.98 ± 0.07 ^a^ (94)	8.54 ± 0.20 ^a^ (82)	3.89 ± 0.11 ^a^ (82)	9.40 ± 0.12 (82)
	GA	2.54 ± 0.15 ^b^ (22)	7.64 ± 0.27 ^b^ (18)	3.30 ± 0.11 ^b^ (18)	9.14 ± 0.25 (18)
*p* value		0.0009	0.0311	0.0031	0.3383

The different superscripts within the same column differ significantly at *p <* 0.05, *p <* 0.01, and *p <* 0.001. The values in the parentheses indicate the number of observations per SNP genotype.

**Table 6 animals-14-01945-t006:** The constructed haplotypes are based on the identified polymorphisms of the *PPARGC1A* gene and their association with the milk traits of the riverine buffalo of Bangladesh.

Haplotype	Observed Frequency	DMY (Liter)	Fat%	Protein%	SNF%
Hap1: CAAG	0.26	2.89 ± 0.08 ^ab^ (93)	8.57 ± 0.18 ^a^ (75)	3.68 ± 0.08 ^ab^ (76)	9.22 ± 0.12 (79)
Hap2: CAGG	0.12	2.96 ± 0.11 ^ab^ (43)	8.63 ± 0.24 ^a^ (34)	3.77 ± 0.13 ^ab^ (34)	9.02 ± 0.16 (35)
Hap3: CGAG	0.18	2.91 ± 0.09 ^ab^ (63)	8.51 ± 0.21 ^a^ (51)	3.78 ± 0.11 ^ab^ (51)	9.15 ± 0.14 (52)
Hap4: CGGG	0.16	3.01 ± 0.10 ^a^ (58)	8.55 ± 0.21 ^a^ (47)	3.91 ± 0.12 ^a^ (48)	9.26 ± 0.16 (49)
Hap5: CAGA	0.02	2.40 ± 0.28 ^bc^ (8)	8.08 ± 0.55 ^ab^ (5)	3.21 ± 0.21 ^b^ (5)	9.01 ± 0.36 (5)
Hap6: CAAA	0.06	2.47 ± 0.67 ^bc^ (19)	7.80 ± 0.30 ^ab^ (15)	3.38 ± 0.10 ^b^ (14)	9.17 ± 0.26 (15)
Hap7: CGAA	0.05	2.46 ± 0.15 ^bc^ (17)	7.41 ± 0.32 ^ab^ (14)	3.34 ± 0.11 ^b^ (14)	9.16 ± 0.21 (14)
Hap8: TGAG	0.06	2.84 ± 0.14 ^abc^ (21)	7.97 ± 0.33 ^ab^ (19)	3.80 ± 0.19 ^ab^ (19)	9.36 ± 0.26 (19)
Hap9: TAAG	0.04	2.65 ± 0.25 ^abc^ (11)	7.95 ± 0.49 ^ab^ (9)	3.91 ± 0.29 ^ab^ (9)	9.48 ± 0.32 (9)
Hap10: TGAA	0.03	2.30 ± 0.22 ^bc^ (9)	6.91 ± 0.31 ^b^ (9)	3.60 ± 0.07 ^ab^ (9)	9.59 ± 0.19 (9)
Hap11: TAAA	0.02	2.01 ± 0.15 ^c^ (7)	7.13 ± 0.37 ^ab^ (7)	3.62 ± 0.08 ^ab^ (7)	9.60 ± 0.25 (7)
Level of Significance		***	***	***	NS

The different superscripts within the same column differ significantly at *p <* 0.001. The values in the parentheses indicate the number of respective haplotypes. ***, level of significance at *p <* 0.001; NS, non-significant.

## Data Availability

The data analyzed in this study are not currently publicly available, as additional analytical studies will be conducted in the future. It can be obtained from the corresponding author upon reasonable request.

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
