# Peer review of "Detection of Polymorphisms in FASN, DGAT1, and PPARGC1A Genes and Their Association with Milk Yield and Composition Traits in River Buffalo of Bangladesh"

_animals, 2024, doi:10.3390/ani14131945_

Round 1
Reviewer 1 Report
Comments and Suggestions for Authors
The author have identified some SNPs associated with milk traits in Bangladesh buffalo population.The results of this manuscript are reliable and clear, but the following questions still need to be addressed:
1. Line199: a pooled DNA samples was used. What is the purpose of this?
2. Line207-209: please rewrite this sentence.
3. How is the haplotype constructed? How is their correlation analysis implemented? The manuscript is not described in the Methods section.
4. How do you perform the SNP genotyping for the population?
Comments on the Quality of English LanguageThe author have identified some SNPs associated with milk traits in Bangladesh buffalo population.The results of this manuscript are reliable and clear, but the following questions still need to be addressed:
1. Line199: a pooled DNA samples was used. What is the purpose of this?
2. Line207-209: please rewrite this sentence.
3. How is the haplotype constructed? How is their correlation analysis implemented? The manuscript is not described in the Methods section.
4. How do you perform the SNP genotyping for the population?
Author Response
Response to Reviewer 1 Comments
The comments by the reviewers were greatly appreciated and the manuscript was revised faithfully accordingly. The corrected sentences are highlighted in red font within the document (animals-3066771.docx). The responses to the reviewer comments were marked in red color in this document.
Comments
The author have identified some SNPs associated with milk traits in Bangladesh buffalo population.The results of this manuscript are reliable and clear, but the following questions still need to be addressed:
Comments 1: Line199: a pooled DNA samples was used. What is the purpose of this?
Response 1: Thank you for your valuable comment. Using pooled DNA samples is a common procedure in DNA-based studies to initially screen for polymorphisms in the investigated populations. Therefore, we initially used pooled DNA samples (three DNA samples combined) to determine whether the selected primers were effective for the Bangladeshi river-type buffalo population.
Comments 2: Line 207-209: please rewrite this sentence.
Response 2: The requested sentence has been revised and can now be found in lines 206-208.
Comments 3: How is the haplotype constructed? How is their correlation analysis implemented? The manuscript is not described in the Methods section.
Response 3: The haplotype construction was performed using DnaSP v.6.12 software. The association analysis was performed in R, considering genotype, location, and management system as fixed effects. This information has now been included in the Methods section on lines 211-213.
Comments 4: How do you perform the SNP genotyping for the population?
Response 4: Initially, we conducted Sanger sequencing on the amplified PCR product. Subsequently, we detected polymorphisms and determined SNP-based genotypes using bioinformatic tools such as Chromas and ClustalW.

Reviewer 2 Report
Comments and Suggestions for Authors
The authors presented a paper on Detection of Polymorphisms in FASN, DGAT1, and PPARGC1A 2 Genes and Their Association with Milk Yield and Composition Traits in River Buffalo of Bangladesh.
General comment: the work is carried out with the approach of the candidate genes. Even if we are in the genomic era, I think that the experimental design is correct and that the work is well done. However, there are some points of concern that make the manuscript not acceptable in the present form.
Specific comments:
Keywords: please try to avoid words already included in the title.
Introduction: to better describe this buffalo please specify also the chromosomes number.
Lines 177-178: please specify the modifications applied to the protocol.
Lines 184, 229, 244, 263: please reorder and renumber the tables since Table 2 is the first to be quoted in the text.
Line 218: did the authors considered to add age of the animal and lactation order as effects in the model? The importance of these effects should not be ignored as also specified in lines 363-364.
Figure 2 could be moved to Supplementary materials.
Lines 270-290: this paragraph could be strongly reduced since most of the information are reported in Table 3.
Line 310: please add the same footnote as in Tables 5 and 6 and specify the number of animals doesn’t match with the previously reported numbers.
Table 5: It is reported a difference in means between the g.409452G>A genotypes for the SNF% trait, but the P value results > 0.05, is it correct?
In the results and discussion paragraph, the significative differences in means between the genotypes should also be addressed and discussed, not only the general association between the polymorphism and the trait. Moreover, on several occasions the significative difference appears only within the minor allele, could the very low sample numbers of the minor allele in those cases play a role in resulting significant? Lastly, has a multiple testing correction been applied, since a lot of analysis have been carried out?
Author Response
Response to Reviewer 2 Comments
The comments by the reviewers were greatly appreciated and the manuscript was revised faithfully accordingly. The corrected sentences are highlighted in red font within the document (animals-3066771.docx). The responses to the reviewer comments were marked in red color in this document.
Comments
The work is carried out with the approach of the candidate genes. Even if we are in the genomic era, I think that the experimental design is correct and that the work is well done. However, there are some points of concern that make the manuscript not acceptable in its present form.
Comments 1: Keywords: please try to avoid words already included in the title.
Response 1: We have taken your suggestion into account and have revised the keywords accordingly on line 58.
Comments 2: Introduction: to better describe this buffalo please specify also the chromosomes number.
Response 2: Thank you for your suggestion. The chromosome number has been included in lines 68-70 of the manuscript.
Comments 3: Lines 177-178: please specify the modifications applied to the protocol.
Response 3: In addition to the manufacturer’s instructions for DNA extraction, we used RBC lysis buffer to break down the red blood cells, which allows for better DNA concentration. We have revised the sentence to include this detailed information in lines 176-177.
Comments 4: Lines 184, 229, 244, 263: please reorder and renumber the tables since Table 2 is the first to be quoted in the text.
Response 4: To ensure the chronological flow of the manuscript, we have revised the order of the tables as suggested.
Comments 5: Line 218: did the authors considered to add age of the animal and lactation order as effects in the model? The importance of these effects should not be ignored as also specified in lines 363-364.
Response 5: We mentioned the age of the animals in lines 157-158. However, smallholder farmers typically do not maintain Herdbook-based record-keeping, making it challenging to accurately determine the animals' ages. Consequently, we were unable to include these two important effects in the model.
Comments 6: Figure 2 could be moved to Supplementary materials.
Response 6: We appreciate the suggestion to move Figure 2 to Supplementary Materials. However, we believe Figure 2 is integral to illustrating key findings and enhancing the manuscript's coherence and clarity. Therefore, we propose retaining Figure 2 in its current position to maintain the manuscript's integrity and effectively support our objectives.
Comments 7: Lines 270-290: this paragraph could be strongly reduced since most of the information are reported in Table 3.
Response 7: Thank you for your valuable comments. We have deleted the redundant portion and revised the section accordingly, now presented in lines 278-281.
Comments 8: Line 310: please add the same footnote as in Tables 5 and 6 and specify the number of animals doesn’t match with the previously reported numbers.
Response 8: As per your recommendation, we have revised this to include the same footnote as in Tables 4, 5 and 6. Upon thorough review, the provided information has been confirmed to be accurate.
Comments 9: Table 5: It is reported a difference in means between the g.409452G>A genotypes for the SNF% trait, but the P value results > 0.05, is it correct?
Response 9: Thank you for your observation. We reviewed our analyzed dataset and identified an error in the interpretation of the SNF mean values in Table 5. We have since revised it accordingly.
Comments 10: In the results and discussion paragraph, the significative differences in means between the genotypes should also be addressed and discussed, not only the general association between the polymorphism and the trait. Moreover, on several occasions the significative difference appears only within the minor allele, could the very low sample numbers of the minor allele in those cases play a role in resulting significant? Lastly, has a multiple testing correction been applied, since a lot of analysis have been carried out?
Response 10: Thank you for your suggestion. In some cases, we did describe and compare results at the SNP level. However, we discussed the findings more generally in most instances because the detected SNPs were not common in other studies. As a result, making comparisons with unmatched polymorphisms is not typical. Therefore, we have maintained our original approach.

Reviewer 3 Report
Comments and Suggestions for Authors
The manuscript aim is to investigate the potential role of SNPs in FASN, DGAT1, PPARGC1A on fatty acids composition of Buffalo milk. Generally speaking, the aim and the results are well pursued and clearly presented along the paper. Discussion section is well built and the conclusion are non-speculative, but based on what was reported in the manuscript. Meanwhile, there are some point to clarify especially on sampling and statistical part.
Key Words:
Please avoid to put in this section words already used in the title.
Introduction:
Line 96-98 need one or more references.
Mat&Met:
As DMY showed a range from 1.03 to 5.50 liters among the sampled animals, it should be clarified the age of animals and the stage of lactation. Moreover, this fixed effects must be included in the linear model.
Results:
As the min and max values of the recorded phenotypes are extreme, but the SD seemed to be regular, maybe a visualization of the data through Violin Plots could give to the reader the idea of the distribution of the phenotypes data.
Comments on the Quality of English Language
English is overall clear
Author Response
Response to Reviewer 3 Comments
The comments by the reviewers were greatly appreciated and the manuscript was revised faithfully accordingly. The corrected sentences are highlighted in red font within the document (animals-3066771.docx). The responses to the reviewer comments were marked in red color in this document.
Comments
The manuscript aim is to investigate the potential role of SNPs in FASN, DGAT1, PPARGC1A on fatty acids composition of Buffalo milk. Generally speaking, the aim and the results are well pursued and clearly presented along the paper. Discussion section is well built and the conclusion are non-speculative, but based on what was reported in the manuscript. Meanwhile, there are some point to clarify especially on sampling and statistical part.
Comments 1: Key Words: Please avoid to put in this section words already used in the title.
Response 1: We have taken your suggestion into account and have revised the keywords accordingly on line 58.
Comments 2: Introduction: Line 96-98 need one or more references.
Response 2: We have addressed this concern by adding the suggested reference to the specified sentence on lines 98-100.
Comments 3: Mat&Met: As DMY showed a range from 1.03 to 5.50 liters among the sampled animals, it should be clarified the age of animals and the stage of lactation. Moreover, this fixed effects must be included in the linear model.
Response 3: Thank you for your insightful comments on our manuscript. We followed ICAR guidelines for test day buffalo milk recording. Subsequently, we calculated both 280-day milk yield and daily milk yield, which mitigates the necessity of considering the lactation stage as fixed effects in our model. However, due to a significant number of missing values for parity numbers, it was not feasible to incorporate parity numbers as a fixed effect in the analysis. We hope these clarifications and adjustments address your concerns adequately.
Comments 4: Results: As the min and max values of the recorded phenotypes are extreme, but the SD seemed to be regular, maybe a visualization of the data through Violin Plots could give to the reader the idea of the distribution of the phenotypes data.
Response 4: Thank you for your thorough review of our manuscript. We appreciate your suggestion regarding the visualization of phenotype data through Violin Plots. In response to your comment, we have included a depiction of the phenotype distribution in supplementary Figure S1 and integrated corresponding text into lines 166-168 of the revised manuscript.

Reviewer 4 Report
Comments and Suggestions for Authors
Simple summary: no Comments
Abstract: no Comments
Introduction: In line 98, one sentence ended mentioning heart-healthy option. There was no reference supporting that sentence.
Last paragraph, line 122-142, can be shorten, bit too long.
Material and methods: Bit too long but no change needed.
Results: no Comments
Discussion: In third paragraph, line 366-383, your result is compared with other published data. I don't see mentioning any work from Indian buffaloes. Are there any reason for that?
Conclusion: no Comments
Author Response
Response to Reviewer 4 Comments
The comments by the reviewers were greatly appreciated and the manuscript was revised faithfully accordingly. The corrected sentences are highlighted in red font within the document (animals-3066771.docx). The responses to the reviewer comments were marked in red color in this document.
Comments 1: Simple summary: no Comments.
Response 1: Thank you for taking the time to review our manuscript and for providing valuable feedback.
Comments 2: Abstract: no Comments
Response 2: Thank you for your feedback on the abstract section of the manuscript.
Comments 3: Introduction: In line 98, one sentence ended mentioning heart-healthy option. There was no reference supporting that sentence.
Response 3: We have incorporated a reference to substantiate the statement in lines 98-100.
Comments 4: Last paragraph, line 122-142, can be shorten, bit too long.
Response 4: The last paragraph has been revised, lines 124-137.
Comments 5: Material and methods: Bit too long but no change needed.
Response 5: Thank you for your feedback on the Material and Methods section. We appreciate your input on this matter.
Comments 6: Discussion: In third paragraph, line 366-383, your result is compared with other published data. I don't see mentioning any work from Indian buffaloes. Are there any reason for that?
Response 6: Thank you for bringing this to our attention. The comparison with other published data was based on a random selection of studies, and we did not specifically exclude works related to Indian buffaloes. We apologize for any oversight in not including relevant studies from this specific context.
Comments 7: Conclusion: no Comments
Response 7: Thank you for your feedback on the conclusion of our manuscript. We appreciate your acknowledgment that no further comments are necessary in this section.

Round 2
Reviewer 1 Report
Comments and Suggestions for Authors
No comments
Comments on the Quality of English LanguageNo comments
Author Response
Response to Reviewer 1 Comments
Comments: No comments
Response: We appreciate your thorough review of our manuscript. We are pleased to note that you have no additional comments. Thank you for your time and consideration.

Reviewer 2 Report
Comments and Suggestions for Authors
First of all I want to thank the authors for their efforts in improving the manuscript that in the present revised version is now more understandable for the readers. I am satisfied with this revised version.
Author Response
Response to Reviewer 2 Comments
Comments: First of all I want to thank the authors for their efforts in improving the manuscript that in the present revised version is now more understandable for the readers. I am satisfied with this revised version.
Response: Thank you for your positive feedback and for acknowledging our efforts in improving the manuscript. We are pleased to hear that the revised version is now more understandable for the readers and that you are satisfied with our revisions. We appreciate your valuable comments and support throughout the review process.
